# Identifying common health indicators from paediatric core outcome sets: a systematic review with narrative synthesis using the WHO International Classification of Functioning, Health and Disability

Victoria Harbottle [1,2] Bronia Arnott,[2] Chris Gale [3] Elizabeth Rowen,[1] Niina Kolehmainen [2]

[1]Rehabilitation Department, Great North Children's Hospital, Newcastle Upon Tyne, UK
[2]Population Health Sciences Institute, Newcastle University, Newcastle upon Tyne, UK
[3]Academic Neonatal Medicine, Imperial College London, London, UK

**Correspondence to**
Victoria Harbottle; Victoria.Harbottle@newcastle.ac.uk

## ABSTRACT

**Background** Indicators of child health have the potential to inform societal conversations, decision-making and prioritisation. Paediatric core outcome sets are an increasingly common way of identifying a minimum set of outcomes for trials within clinical groups. Exploring commonality across existing sets may give insight into universally important and inclusive child health indicators.

**Methods** A search of the Core Outcome Measures in Effectiveness Trial register from 2008 to 2022 was carried out. Eligible articles were those reporting on core outcome sets focused on children and young people aged 0–18 years old. The International Classification of Functioning, Disability and Health (ICF) was used as a framework to categorise extracted outcomes. Information about the involvement of children, young people and their families in the development of sets was also extracted.

**Results** 206 articles were identified, of which 36 were included. 441 unique outcomes were extracted, mapping to 22 outcome clusters present across multiple sets. Medical diagnostic outcomes were the biggest cluster, followed by pain, communication and social interaction, mobility, self-care and school. Children and young people's views were under-represented across core outcome sets, with only 36% of reviewed studies including them at any stage of development.

**Conclusions** Existing paediatric core outcome sets show overlap in key outcomes, suggesting the potential for generic child health measurement frameworks. It is unclear whether existing sets best reflect health dimensions important to children and young people, and there is a need for better child and young person involvement in health indicator development to address this.

## INTRODUCTION

*Measure what is measurable, and make measurable what is not. (Isaac Newton)*

How society measures characteristics of health is important because the act of measuring is an act of making an aspect

### WHAT IS ALREADY KNOWN ON THIS TOPIC

⇒ There are aspects of children and young people's health that are not routinely, universally measured.
⇒ This lack of data about children and young people's health can hinder health policy, care and research.
⇒ Core outcome sets are a way of providing guidance on what to measure, but often relate to specific clinical populations only.

### WHAT THIS STUDY HOPES TO ADD

⇒ This study identifies common, universally shared themes across core outcome sets.
⇒ These themes help identify key universal indicators of child health that should be measured across populations.
⇒ The study further identifies gaps and areas for improvement in the involvement of children and their families in identifying key health indicators.

of health visible—in societal discussion, decision-making and resource allocation. Measurable health characteristics are often referred to as indicators, and how they are defined for measurement shapes the information that is collected and available for decisions. Measuring health through routine and other large-scale data sets is increasingly common, and the use of the resulting data in societal conversations, decision-making and prioritisation likely has major consequences for people's lives.

Many important aspects of child health are not yet routinely measured. While some health indicators such as births, deaths and vaccinations are long-established, many others lack robust data. The COVID-19 pandemic provides an illustrative example, where the absence of

data about children's physical and mental health and development has hindered analyses of the impacts of the pandemic on children and, arguably, limited families', decision-makers' and professionals' ability to articulate their concerns. This, in turn, has hindered effective decision-making about important policy decisions such as opening of schools. The need to rethink the measurement of child health is gaining policy traction internationally[1–3] and the choice over what indicators to measure is now paramount.[4]

Measuring child health is hindered by lack of agreement about the important, universally applicable and clinically meaningful indicators of child health and well-being. The past decade has seen a positive, increased focus on common research outcomes relevant across professional disciplines, illustrated for example in the development of core outcome sets. Core outcome sets provide an agreed, standardised minimum set of outcomes to be reported for a specific clinical group in clinical research and increasingly in clinical practice.[5] There is no standardised methodology for core outcome set development, but they generally involve stakeholders agreeing the most important outcomes to report using consensus methodology.[6] However, although patient involvement in core outcome set development has been recognised important,[7] the sets continue to be fundamentally organised around clinical and diagnostic groups to service healthcare and trials. They do not currently provide an off-the-shelf set for universally important indicators of child health that could be applied across diagnostic categories, and across healthy and clinical populations.

We believe there is a need for a generic, universal, inclusive set of core child health and well-being indicators, and we think such a set should: (1) take the child's perspective (as opposed to, for example, medical or educational); (2) adopt a life-course view, that is, consider future health and well-being prospects as well as present health; (3) be practically oriented, with real potential to inform routine data collection and societal discourse, policy and interventions; and (4) be inclusive of diverse health and developmental trajectories.

As the first step, in 2017–2020, we engaged with young people, families and decision-makers to discuss broad ideas about health indicators that might matter to them. Informed by those discussions, the present study sought to identify common themes in existing paediatric core outcome sets, as a way to contribute to the wider efforts to progress towards a generic, universal, inclusive set of core child health and well-being indicators. The present paper reports on two specific objectives: (1) to identify common indicators of health included in published paediatric core outcome sets for children and youth (0–18 years); and (2) to explore how children, young people and their families' perspectives informed the selection of these outcomes.

## METHODS

This study used established review methods[8 9] to identify and select papers reporting on published paediatric core outcome sets. Narrative synthesis was used to analyse the data.[10] A protocol was agreed by the review team, with objectives, selection criteria and data extraction techniques agreed in advance. The results are reported in accordance with the Preferred Reporting Items for Systematic Reviews and Meta Analyses 2020 statement.[11] The review objectives fell outside the scope of the International Prospective Register of Systematic Reviews.[12] We used the WHO International Classification of Functioning, Disability and Health (ICF),[13] an existing consensus framework of health. The ICF is an internationally recognised set of domains of health and functioning that provides a structured way to understand and classify an individual's health and functioning. It considers health indicators relating to body functions and structures, activities and participation and provides multidisciplinary terminology to classify them.[14]

### Search

Paediatric core outcome sets were identified from the Core Outcome Measures in Effectiveness Trial (COMET) register.[15] The register is the internationally recognised database of ongoing and completed studies relating to core outcome set development. It is updated by systematic review annually, and by authors registering their sets on an ad hoc basis. The last review carried out by COMET, identifying new published core outcome sets for addition to the database was published in January 2021 including studies up to and including December 2018.[16] To identify paediatric sets, we systematically searched the COMET register[15] from January 2008 to March 2022 using the terms indicated in box 1. The search was updated to incorporate the COMET database review up to and including December 2019.[17]

### Selection

Articles were eligible for inclusion in this review if: (1) they reported on a disease or diagnostic group-specific core outcome sets developed using consensus methods (defined as reporting a clear criterion for determining outcome inclusion/exclusion, the number of people in each stakeholder group at each stage, and all outcome scoring[7]; (2) focused on children and young people (0–18 years) as the health beneficiaries; and (3) was published after 2008 (a previous systematic review[18] identifies work prior to this date). Systematic reviews, core outcome set protocols and core outcome sets without reported consensus methods were excluded, as were articles reporting core outcome sets spanning children and adults. Titles and abstracts were first independently reviewed by one author (VH) with a randomly selected

25% double screened by a second author (ER). Full texts of articles meeting the inclusion criteria were then further reviewed, with a randomly selected 50% double screened. Reasons for exclusion were recorded at both stages. Agreement between authors at screening and full-text stages was 94% and 82%, respectively, and discrepancies were resolved through discussion.

### Data extraction and analysis

Study-specific forms were developed, and piloted; and subsequently used by two authors (VH, ER) to independently extract data on: authorship; the core outcome set scope, use and outcomes listed; and consensus methodology used, including method type, stakeholder numbers and patient/public involvement. The ICF was used as a framework for coding the outcome data; this provided a common, internationally agreed terminology for naming and describing health outcomes. Outcomes were considered health indicators if they directly related to the child. Outcomes related to care inputs and processes (eg, resources, length of stay, attendance) were excluded. Outcomes were coded using a published ICF decision tree.[14] For the full mapping, please see the full data set.[19] Article screening was done in EndNote; data extraction in Microsoft Word; and analysis in Microsoft Excel.

## RESULTS

We identified 206 records, of which 68 duplicates were removed. A further 86 records were excluded following title and abstract screening, 16 were excluded following full-text assessment (figure 1) and 36 studies were selected for inclusion in full review (table 1). Selected articles described 36 core outcome sets related to:

gastrointestinal conditions (n=8); neurological conditions (n=7); ear, nose and throat (n=5); orthopaedics (n=4); general paediatrics (n=3); neonatology (n=2); respiratory (n=2); metabolic disease (n=2); and rheumatology, oncology and dentistry specialities (n=1 each) (table 1). The number of outcomes in a core set ranged from 3 to 39, with a median of 9.

### Common outcome domains

From the 36 core outcome sets included, 441 outcomes were extracted. Mapping these to the ICF resulted in 22 clusters of outcomes, linked to 25 unique ICF codes (table 2). Medical diagnoses formed the largest cluster, activity and participation forming the majority of the other larger clusters. Several smaller clusters related to body functions were identified as well as a cluster relating to growth. Personal factors relating to emotional well-being were clustered and showed relatively higher commonality. Environmental factors were included in several sets and collectively formed one, large cluster. table 3 summarises the top 10 clusters by size after medical diagnostic outcomes and shows how many sets each outcome cluster was represented in.

### Stakeholder involvement

Table 4 summarises the stakeholder involvement in the three main sections of core outcome set development: (1) generation of an outcome longlist; (2) consensus process; and (3) final consensus meeting. The diversity of stakeholders involved varied (table 4). Five (14%, 5/36) only sought opinion from clinicians/researchers throughout their development, including no parents or children or young people (CYP) at any stage. The remaining 31 (86%, 31/36) included parents/caregivers in at least 1 part of the development, with 16 including parents/

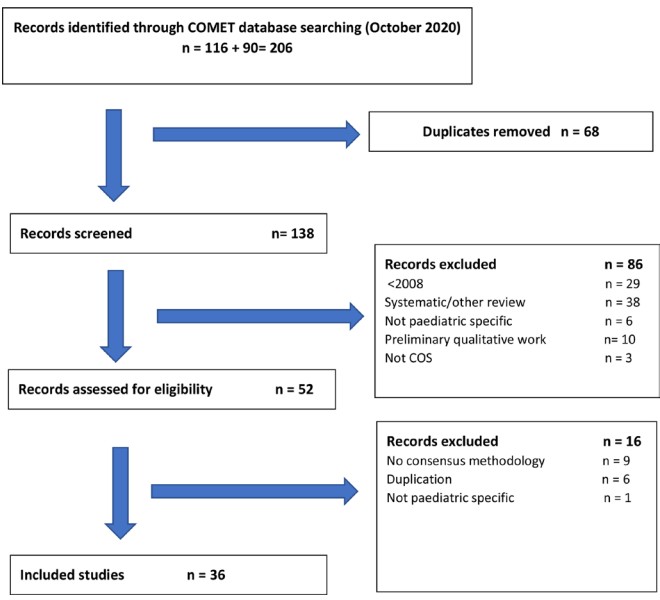

**Figure 1** Summary of screening strategy. COMET- Core Outcome Measures in Effectiveness Trial. COS- Core Outcome Set

| Box 1 | COMET Database Search Criteria |
| --- | --- |

Health Area-disease category
Child Health
AND
Publication year
2008 onwards
AND
Published/unpublished
Published

**OR**
Minimum age
0
AND
Maximum age
18
AND
Publication year
2008 onwards
AND
Published/unpublished
Published

**Table 1** Overview of included paediatric core outcome sets

| Author | Scope of set | | | Use | Size of set |
| | Age range | Condition area | Condition | | |
|---|---|---|---|---|---|
| Alin et al[26] | Birth upwards | Gastrointestinal | Gastroschisis | Research | 8 |
| Rexwinkle et al[27] | 1–18 years | Gastrointestinal | Gastro-oesophageal refulx | Research and clinical guidelines | 9 |
| Sherratt et al[20] | <18 years | Gastrointestinal | Uncomplicated appendicitis | Research | 14 |
| Singendonk et al[23] | 'Infants' | Gastrointestinal | Infant gastro-oesophageal reflux disease | Research | 9 |
| Steutel et al[28] | 'Infants' | Gastrointestinal | Infant colic | Research | 6 |
| Karas et al[29] | 'Paediatric' | Gastrointestinal | Acute diarrhoea | Research | 5 |
| Kuizenga-Wessel et al[30] | 0–18 years | Gastrointestinal | Functional constipation | Research | 8 |
| Zeevenhooven et al[25] | 'Paediatric' | Gastrointestinal | Functional abdominal pain disorders | Research | 8 |
| Crudgington et al[31] | 5–16 years | Neurology | Rolandic epilepsy | Research but mentions clinical use (audit, SE) | 39 |
| Joachim et al[32] | 'Children' | Neurology | Neurological impairment and enteral tube feeding | Research | 12 |
| Murugupillai et al[33] | Two sets: Pre-school<6 years School-age 6–18 years | Neurology | Effectiveness of antiepilepsy therapy in children | Research | 8 |
| Morris et al[4] | 'Children' | Neurology | Neurodisability | Informing development of NHS Outcomes Framework | 12 |
| Nabbout et al[34] | 2–18 years | Neurology | Dravet syndrome | Research | 5 |
| Pease et al[35] | 0–18 years | Neurology | Paediatric cerebral visual impairment | Research | 27 |
| Butler et al[36] | 0–18 years | Neurology | Facial palsy | Quality of care | 20 |
| Allori et al[37] | 'Child' | Ear, nose and throat | Cleft lip/palate | Research+clinical (benchmarking) | 22 |
| Balakrishnan et al[38] | 'Paediatric' | Ear, nose and throat | Head and neck lymphatic malformations | Research | 31 |
| Balakrishnan et al[39] | 'Paediatric' | Ear, nose and throat | Laryngotracheal reconstruction | Research and clinical | 8 |
| Harman et al[40] | <18 years | Ear, nose and throat | Otitis media with effusion (in cleft palate) | Research | 9 |
| Liu et al[41] | <12 years | Ear, nose and throat | Otitis media with effusion (in otherwise healthy children) | Research | 9 |
| Leo et al[42] | 'Children' | Orthopaedics | Perthes disease | Research and clinical practice | 14 |
| Marson et al[43] | 5–16 years | Orthopaedics | Limb fractures | Research | 8+1 additional for upper or lower limb |
| De Graaf et al[44] | 'Children' | Orthopaedics | Acute osteomyelitis and septic arthritis | Research-specific RCT being planned | 11 |
| Pondaag and Malessy[45] | Not stated | Orthopaedics | Brachial plexus birth injury | Research and clinical– universal dataset | 3 |
| Matvienko-Sikar et al[46] | <1 year | General paediatrics | Infant feeding for the prevention of childhood obesity | Research | 25 |
| Palermo et al[47] | 'Paediatric' | General paediatrics | Chronic pain | Research | 4 mandatory, 3 optional |
| Topjian et al[48] | 'Children' | General paediatrics | Cardiac arrest | Research | 5 |
| Bösch et al[49] | 'Paediatric' | Metabolic disease | Intoxication-type inborn errors of metabolism | Research | 17 |
| Pugliese et al[50] | <12 years | Metabolic disease | Medium-chain acyl-coenzyme A dehydrogenase deficiency (MCAD) and phenylketonuria (PKU) | Research | MCAD- 8 PKU- 9 |
| Damhuis et al[51] | Birth | Neonatology | Growth restriction in newborns | Research | 19 |
| Webbe et al[52] | Neonates | Neonatology | Neonatology | Research | 12 |
| Gilchrist et al[53] | 'Children' | Respiratory | Protracted bacterial bronchitis | Research | 6 |
| Sinha et al[24] | <5 years and 5–18 considered separately | Respiratory | Chronic childhood asthma | Research, pilot | 4 |
| Haeusler et al[54] | 'Children' | Oncology | Fever and neutropenia | Research | 10 |
| Heiligenhaus et al[55] | Implied <16 years | Rheumatology | Juvenile idiopathic arthrtis-associated uveitis | Research | 7 |
| Smail-Faugeron et al[56] | Children with primary teeth | Dental | Pulp treatment for primary teeth | Research | 5 |

**Table 2** Overview of coded and grouped outcomes extracted from paediatric core outcome sets

| Cluster (n total) | ICF code | n | Examples |
|---|---|---|---|
| Communication (n=18) | Voice and speech functions (b3) | 7 | Articulation, overall speech, speech ability |
| | Communication (d3) | 11 | Receptive language skills, listening skills, receptive communication |
| Self-care (n=18) | Eating (d550) Drinking (including breast feeding) (d560) | 9 | Breast feeding, child self/assisted feeding, feeding difficulties |
| | Self-care (d5) | 9 | Activities of daily living, toileting, safety |
| Mobility, movement and related structures (n=19) | Mobility (d4) | 10 | Gross motor, fine motor, motor impairment, mobility |
| | Mobility of joint functions (b710) | 4 | Hip mobility, passive range of movement, active range of movement |
| | Structures relating to movement (s7) | 5 | Limb deformity, femoral head shape, arthritic changes to hip |
| School and learning (n=19) | School education (d820) | 11 | School attendance, engagement in school life, |
| | Learning and applying knowledge (d1) | 8 | Literacy, academic attainment, school performance |
| Social (n=18) | Interpersonal interactions and relationships (d7) | 11 | Friendships, relationship with parents/siblings, psychosocial development |
| | Managing own behaviour (d520) | 7 | Behavioural concerns, behaviour |
| Community, play and civic life (n=14) | Community, social and civic life (d9) | 14 | Sport participation, time away from activities, social activities, play |
| Growth (n=12) | Growth maintenance functions (b560) | 6 | Head circumference, length, adequate growth |
| | Weight maintenance functions (b530) | 6 | Weight, weight gain over time, body composition |
| Pain (n=17) | Pain (b280–b289) | 17 | Pain, abdominal pain, |
| Mental functions (n=11) | General mental functions (b1) | 5 | General cognitive ability, cognitive impairment, psychosocial development |
| | Specific mental functions (b140–199) | 6 | Memory, executive function |
| Sleep (n=11) | Sleep (b134) | 11 | Duration of sleep, awakenings from sleep, sleep quality |
| Function of senses (n=11) | Hearing functions (b230) | 4 | Hearing, hearing impairment, |
| | Seeing functions (b210) | 7 | Visual acuity, visual performance |
| Digestive system functions (n=8) | Digestive functions (b515) | 4 | Bowel obstruction, vomiting |
| | Defecation functions (b525) | 4 | Defecation pattern, stool consistency, defecation frequency |
| Respiratory functions (n=4) | Respiration functions (b440) | 3 | Breathing difficulties, airway compromise, airway obstruction |
| | Additional respiratory functions (b450) | 1 | Cough |
| Skin (n=2) | Structure of areas of skin (s810) | 2 | Skin involvement, skin necrosis |
| Environmental factors (n=33) | e | 33 | Family life, mother's knowledge of how to offer food, family stress, family quality of life |
| Medical diagnosis (n=84) | N/A | 84 | Sepsis, liver disease, infection, seizure |
| Quality of life (n=19) | N/A | 19 | |
| Death (n=13) | N/A | 13 | Death, mortality, all-cause mortality, infection-related mortality |
| Ear, nose, throat and dental (n=12) | N/A | 12 | Dental health, oral health, occlusion, facial profile, smile, pathologic root resorption |
| Personal factors relating to emotional well-being (n=17) | N/A | 17 | Feelings, self-esteem, feelings about having epilepsy, fear of seizures, emotional well-being |
| Other personal factors (n=10) | N/A | 10 | Concealment of condition, psychosocial development, attitude towards disease |

caregivers in both outcome listing and consensus stages. Overall, 11 (31%, 11/36) included CYP in at least 1 part, with 9 (25%, 9/36) including them in both outcome listing and consensus (table 4). Seven sets related to neonatal or infant populations; three of these were for conditions likely to impact into later childhood (gastroschisis, brachial plexus injury, infants receiving neonatal care) and so could have considered including older CYP. Parent/caregiver and young person stakeholder representation had increased over time (figure 2).

The median number of rounds used for consensus methodologies was 2 (range 1–7), with only 3 studies (8%, 3/36) using only one consensus round. Across the remaining 33 studies using multiple rounds, attrition rates of included stakeholders varied with a mean of 21% (range 0%–63%) for clinicians, 29% (range 0%–95%) for studies including parents in more than one round and 19% (range 0%–73%) for studies including young people in more than one round. Seven (19%, 7/36) mitigated concerns around attrition by recruiting new parent or CYP stakeholder groups for subsequent

**Table 3** Top 10 outcome clusters by size (excluding medical diagnostic outcomes)

| Cluster | Number of outcomes mapped to cluster (n/441) | Number of sets represented in (n/n, %) |
|---|---|---|
| Environmental | 33 | 10/36, 28% |
| School and learning | 19 | 11/38, 31% |
| Mobility, movement, and related structures | 19 | 9/36, 25% |
| Quality of life | 19 | 19/36, 53% |
| Communication | 18 | 9/36, 25% |
| Self-care | 18 | 11/36, 31% |
| Social | 18 | 7/36, 19% |
| Pain | 17 | 12/36, 33% |
| Personal factors relating to emotional well-being | 17 | 11/36, 31% |
| Community, play and civic life | 14 | 11/36, 31% |

rounds. Patient and public involvement (PPI) in study design was not universally included; only fourteen (39%, 14/36) included parents or young people, either through inclusion in the study steering group, through consultation with an advisory group or through the piloting of methods. There was some evidence that the use of PPI in designing study methods impacted consensus attrition rates, with mean stakeholder attrition lower for studies that utilised PPI (figure 3, table 5).

Figure 4 shows all 36 core outcome sets, plotted by stakeholder (parents and/or children) involvement in longlisting (x-axis) and the consensus process (y-axis). This figure highlights that the sets with greater stakeholder involvement tended to be those including a smaller number of domains. The bubble sizes represent the number of individual outcome domains included in the final core outcome set. Sets nearest the top right-hand corner are those with higher proportions of involvement at both stages of the process (representing higher proportions of stakeholder involvement). Clustered around zero are those sets with very minimal, or no stakeholder involvement at either stage, and those to the left of the plot represent those that used reviews only for the longlisting stage, or for which data around stakeholder participation was unavailable.

## DISCUSSION

We extracted 441 outcome variables from 36 paediatric core outcome sets, and mapped them to 22 outcome clusters of the ICF as well as on environmental factors potentially affecting these outcomes. These 22 areas represent potential child health indicators for measurement in routine and large-scale data. The most common indicator cluster was a child's diagnosis (in 76% of sets). The second most common was pain (in 33% of sets), followed by activity and participation indicators related to self-care, school, personal well-being, community and civic life (all in 31% of sets), communication and social

interactions and mobility (both in 25% of sets). These were followed by body structure and function indicators relating to sleep (22%), mental functions (16%) and growth (in 11% of sets). In addition, three categories of common indicators residing outside the ICF were identified: mortality (in 33% of sets), dental/oral health (in 8%) and quality of life (in 53% of sets). Overall, while there is an in-principle commitment to patient involvement in core outcome set development, the selection and prioritisation of indicators in the included sets were more informed by clinicians and researchers than children, young people or parents who were under-represented.

The present review used a robust search and data extraction strategy, independent double screening and data extraction by two authors, and a published decision tree for the coding of data on the ICF. The search was limited to the COMET initiative registry, which itself is rigorously updated annually, with the update published as a peer-reviewed systematic review. We can therefore be relatively certain that the present study successfully identified articles up to and including the last update of the registry (December 2019); the inclusion after that relies on ad hoc author registration and so it is possible that later sets were not included. We did not assess the quality of the paediatric core outcome sets as this was not one of the aims on this review.

There were some limitations to this work. The search excluded core outcome sets that spanned both children and adults. This was as the focus was on transdiagnostic health indicators specific to children and young people. The priorities, life experiences and opinions of CYP differ compared with adults[20] and we felt including studies with a broader age range would have added adult focused outcomes that were inconsistent with our aims. A further limitation was that while independent double screening and data extraction was undertaken by two authors, only a randomly selected proportion was reviewed by the second author.

The outcome clusters identified in the present review align with other, concurrent initiatives. To date, two approaches to selecting indicators and outcomes have dominated: the public health, and the clinical. In the public health approach, coarse indicators (eg, births, deaths, vaccination rates) are selected and used for national and regional reporting and comparison. One such prominent example is the State of Child Health by Royal College of Paediatrics and Child Health.[2] This uses key child health indicators to monitor trends and provide policy recommendation across the UK. Mortality, weight, oral health and mental health are all included, corresponding to some of the domains in the present review. Another example is the Public Health England (PHE) child health indicators, used to monitor trends and inform policy.[21] A challenge with the current public health indicators and related data are children at highest risk of long-term ill health are not well represented—including children with health conditions or marginalised due to sociodemographic circumstances. They are missed

**Table 4** Summary of stakeholder group involvement in paediatric core outcome set development

| Author | Outcome listing | Stakeholders included (numbers) | | | | Consensus method | Stakeholders included (numbers participating across all rounds) | | | | Final consensus meeting | | Steering group explicitly mentioned | | |
| --- | --- | --- | --- | --- | --- | --- | --- | --- | --- | --- | --- | --- | --- | --- | --- |
| | | Clinicians | Parents | Young People | Other | | Clinicians | Parents | Young People | Other | Y/N | P/CYP Included? | Y/N | P/CYP Included? | Other Groups |
| Alin et al[26] | Systematic review | – | – | – | – | Delphi | 49 | 22 | 0 | 0 | √ | Parents only | √ | N | |
| Allori et al[37] | Systematic review, qualitative (clinicians only)—interviews, details not published | – | – | – | – | Modified Delphi | 28 | 2 | 1 | 0 | X | – | √ | √ | X |
| Balakrishnan et al[38] | Survey | 9 | 0 | 0 | 0 | Modified Delphi | 9 | 0 | 0 | 0 | X | – | X | – | X |
| Balakrishnan et al[39] | Survey | 33 | 0 | 0 | 0 | Delphi | 33 | 0 | 0 | 0 | X | – | X | – | X |
| Bösch et al[49] | Systematic review, qualitative focus groups | 0 | 26 | 19 | 0 | Survey (2 rounds) | 30 | 24 | 9 | 0 | √ | √ | X | – | Survey piloted with stakeholders including CYP |
| Butler et al[36] | Review, communication with patient advisory groups | 18 | 3 | 0 | 0 | Modified Delphi | 18 | 3 | 0 | 0 | X | – | √ | X | Patient advisory groups consulted-adults with paediatric-onset palsy |
| Crudgington et al[31] | Systematic review | – | – | – | – | Delphi | 61 | 16 | 3 | 0 | √ | √ | X | X | Consulted young person's advisory group |
| Damhuis et al[51] | Systematic review | – | – | – | – | Delphi | 33 | 14 | 0 | 0 | √ | Parents only | √ | √ | X |
| De Graaf et al[44] | Systematic review | – | – | – | – | Delphi | 97 | 0 | 0 | 0 | √ | Parents+young person representative | X | – | X |
| Gilchrist et al[53] | Systematic review, qualitative interviews with parents, clinician survey | 20 | 16 | 0 | 0 | Delphi | 51 | 0 | 0 | 0 | √ | Parents only | X | – | X |
| Haeusler et al[54] | Review of guidelines and consensus statements | – | – | – | – | Delphi | 33 | 3 | 0 | 0 | X | – | √ | X | X |
| Harman et al[40] | Systematic review, qualitative interviews | – | 43 | 22 | – | Modified Delphi | 75 | 35 | 8 | 8 | √ | X | √ | √ | Study advisory group- invited 1× CYP but withdrew Sought advice from diagnosis specific YPAG |
| Heiligenhaus et al[55] | Systematic review | – | – | – | – | Delphi, nominal group | 16 | 0 | 0 | 0 | √ | X | X | – | X |
| Joachim et al[32] | Systematic review | – | – | – | – | Delphi | 15 | 3 | 0 | 0 | √ | Invited, unable to attend, further remote validation with parents | √ | X | X |
| Karas et al[29] | Systematic review, survey | 66 | 31 | 0 | 4 | Delphi | 64 | 32 | 0 | 0 | X | – | √ | X | X |
| Kuizenga-Wessel et al[30] | Survey | 109 | 165 | 50 | | Modified Delphi | 50 | 80 | 50 | 0 | √ | X | X | X | X |
| Leo et al[42] | Systematic review, qualitative interviews, | 0 | 18 | 12 | 0 | Delphi | 36 | 46 | 18 | 0 | √ | Parents only | X | – | X |

Continued

**Table 4** Continued

| Author | Outcome listing | Stakeholders included (numbers) | | | | Consensus method | Stakeholders included (numbers participating across all rounds) | | | | Final consensus meeting | | Steering group explicitly mentioned | | |
|---|---|---|---|---|---|---|---|---|---|---|---|---|---|---|---|
| | | Clinicians | Parents | Young People | Other | | Clinicians | Parents | Young People | Other | Y/N | P/CYP Included? | Y/N | P/CYP Included? | Other Groups |
| Liu et al[41] | Systematic review, qualitative focus groups, examination of tools used in trials currently | 0 | 3 | 0 | 0 | Survey | 81 | 53 | 0 | 0 | X | – | X | – | X |
| Marson et al[43] | Systematic review, qualitative interviews | 0 | 20 | 20 | 0 | Delphi | 111 | 19 | 0 | 15 | ✓ | ✓ | X | – | Patient, parent and public advisory group consulted |
| Matvienko-Sikar et al[46] | Systematic review and stakeholder meeting | 12 | 0 | 0 | 0 | Delphi | 75 | 2 | 0 | 2 | ✓ | Parents only | X | – | X |
| Morris et al[4] | Systematic review, qualitative focus groups and interviews, Delphi clinicians | 191 | 53 | 54 | 0 | Nominal group | 7 | 5 | 3 | 0 | X | – | ✓ | ✓ | Pilot to refine young person participation methods |
| Murugupillai et al[33] | Survey | 32 | 50 | 15 | 0 | Delphi | 29 | 42 | 0 | 0 | X | – | X | – | X |
| Nabbout et al[34] | Qualitative semistructured interviews informed by systematic review | 4 | 7 | 0 | 0 | Modified Delphi | 7 | 0 | 0 | 0 | X | – | X | – | X |
| Palermo et al[47] | Survey | 52 | 90 | 93 | 0 | Modified Delphi | 44 | 85 | 86 | 0 | ✓ | X | ✓ | X | Patient advisory group consulted |
| Pease et al[35] | Review, qualitative interviews | 0 | 18 | 6 | 0 | Delphi | 52 | 28 | 0 | 0 | ✓ | Parents only | X | – | 2× patient involvement meetings with families, parent and CYP involvement in study design |
| Pondaag and Malessy[45] | Systematic review then survey—binary yes/no for inclusion | 69 | 0 | 0 | 0 | Survey | 59 | 0 | 0 | 0 | X | – | X | – | X |
| Pugliese et al[50] | Systematic review | – | – | – | – | Delphi | 16 | 37 | 0 | 0 | ✓ | Parents only | X | – | 2 patient partner investigators with lived experience, led family advisory forum |
| Rexwinkle et al[27] | Systematic review, health professionals survey and parent/CYP questionnaire | 118 | 146 | 69 | 0 | Delphi | 80 | 130 | 77 | 0 | ✓ | X | X | – | X |
| Sherratt et al[20] | Systematic review+qualitative interviews (parents and CYP) (interviews embedded in unpublished study so no number available) | – | – | – | – | Delphi | 110 | 32 | 3 | 0 | ✓ | ✓ | X | – | Parents and young persons study advisory group |
| Singendonk et al[23] | Survey | 125 | 139 | 0 | 0 | Delphi | 89 | 127 | 0 | 0 | ✓ | X | X | – | X |

Continued

Table 4  Continued

| Author | Outcome listing | Clinicians | Parents | Young People | Other | Consensus method | Clinicians | Parents | Young People | Other | Y/N | P/CYP Included? | Y/N | P/CYP Included? | Other Groups |
|---|---|---|---|---|---|---|---|---|---|---|---|---|---|---|---|
| | | **Stakeholders included (numbers)** | | | | | **Stakeholders included (numbers participating across all rounds)** | | | | **Final consensus meeting** | | **Steering group explicitly mentioned** | | |
| Sinha et al[24] | Survey | 46 | 28 | 11 | 0 | Delphi–clinicians Survey–parents and CYP | 43 | 50 | 0 | 0 | X | – | X | – | Pilot phase tested study materials with parents and young people |
| Smail-faugeron et al[56] | Systematic review+stakeholder meeting | 6 | 0 | 0 | 0 | Delphi | 52 | 0 | 0 | 0 | X | – | X | – | X |
| Steutel et al[28] | Systematic review, Survey | 133 | 55 | 0 | 0 | Delphi | 54 | 43 | 0 | 0 | √ | X | √ | X | X |
| Topjian et al[48] | Used adult set, + suggestions from steering committee | 18 | 0 | 0 | 0 | Delphi | 68 | 6 | 0 | 0 | X | – | √ | X | X |
| Webbe et al[52] | Systematic review × 2 (trials and qual) | – | – | – | – | Delphi | 130 | 53 | 0 | 0 | √ | √ | √ | √ | X |
| Zeevenhooven et al[25] | Systematic review+survey | 152 | 103 | 50 | 0 | Delphi | 104 | 102 | 53 | 0 | √ | X | X | X | X |

CYP, children or young people.

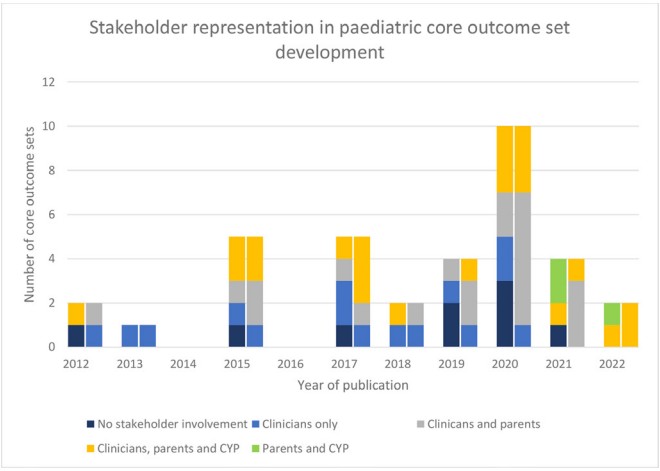

**Figure 2**  Stakeholder groups involved in (1) outcome long listing and (2) consensus process by year. Left hand bar represents outcome long listing, right hand bar represents consensus process. CYP- children or young people.

from data collection, or the data does not cover key indicators relevant to them, for example, the PHE data does not include pain or sleep. In contrast, in the clinical approach, highly specialised end points (eg, particular illness symptoms, treatment side effects) are selected on the basis of their relevance to specific interventions and clinical groups; these indicators and measurement rarely touch on all children universally. There may be a tacit assumption that missing the highest risk children and young people from the public health data is compensated for by the clinical data. However, this is problematic as it neglects some universally important aspects of child health for the clinical populations, and creates a two-strand system where children are viewed through a binary 'typical' versus 'clinical' lens which is then translated into segregated policy and decision-making. For example, the UK Chief Medical Officers developed separate physical activity guidelines for typically developing and disabled children, in large part based on an argument that there

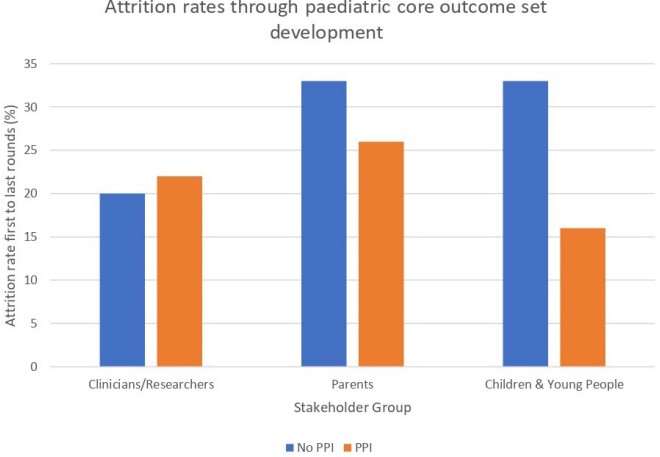

**Figure 3**  Mean attrition between consensus rounds by stakeholder group, comparing studies including patient and public involvement (PPI) in study design to those without.

**Table 5** Attrition of different stakeholder groups throughout consensus rounds

| Author/year | Rounds, n | Attrition first to last rounds (%) | | | Notes |
|---|---|---|---|---|---|
| | | **Clinicians** | **Parents** | **Young people** | |
| Alin et al[26] | 3 | 18 | 48 | – | Attempt to recruit young people, unsuccessful |
| Allori et al[37] | 7 | 0 | 0 | 0 | |
| Balakrishnan et al[38] | 3 | 33 | – | – | |
| Balakrishnan et al[39] | 2 | 0 | – | – | |
| Bösch et al[49] | 2 | 17 | 11 | 0 | |
| Butler et al[36] | 7 teleconferences, each followed by a 2 round Delphi | Attrition rates not recorded | | | |
| Crudgington et al[31] | 2 | 20 | 30 | 0 | |
| Damhuis et al[51] | 3 | 3 | 26 | – | |
| De Graaf et al[44] | 2 | 33 | – | – | |
| Gilchrist et al[53] | 2 | 22 | – | – | |
| Haeusler et al[54] | 4 | 15 | 25 | – | |
| Harman et al[40] | 3 | 30 | – | – | Parents and young people completed 1 round survey only |
| Heiligenhaus et al[55] | 2 (1×Delphi, 1×nominal group | 12 | – | – | |
| Joachim et al[32] | 1 | – | – | – | |
| Karas et al[29] | 2 | 9 | – | – | Different group of parents invited to round 2 |
| Kuizenga-Wessel et al[30] | 2 | – | – | – | Different stakeholder groups of invited to round 2 |
| Leo et al[42] | 2 | 22 | 49 | 33 | |
| Liu et al[41] | 1 | – | – | – | |
| Marson et al[43] | 3 | 31 | 27 | – | |
| Matvienko-Sikar et al[46] | 3 | 63 | 95 | 0 | |
| Morris et al[4] | 1 | – | – | – | 1 round nominal group |
| Murugupillai et al[33] | 2 | 9 | 16 | – | Young people completed 1 round survey only |
| Nabbout et al[34] | 2 | 13 | – | – | |
| Palermo et al[47] | 2 | 15 | 6 | 8 | |
| Pease et al[35] | 2 | 35 | 30 | – | |
| Pondaag and Malessy[45] | 3 | 14 | – | – | |
| Pugliese et al[50] | 3 | 48 | 38 | – | |
| Rexwinkle et al[27] | 1 | – | – | – | |
| Sherratt et al[20] | 3 | 30 | 44 | 73 | |
| Singendonk et al[23] | 2 | – | – | – | Further clinician recruitment for round 2. Different group of parents invited to round 2 |
| Sinha et al[24] | 2 | 6 | – | – | Different group of parents invited to round 2, no young people invited to round 2 |
| Smaïl-Faugeron et al[56] | 3 | 16 | – | – | |
| Steutel et al[28] | 2 | 59 | – | – | Different group of parents invited to round 2 |
| Topjian et al[48] | 2 | 18 | 0 | – | |
| Webbe et al[52] | 3 | 13 | 52 | – | Parents and young people combined in one group |
| Zeevenhooven et al[25] | 2 | 32 | – | – | Different group of parents and young people invited to round 2 |

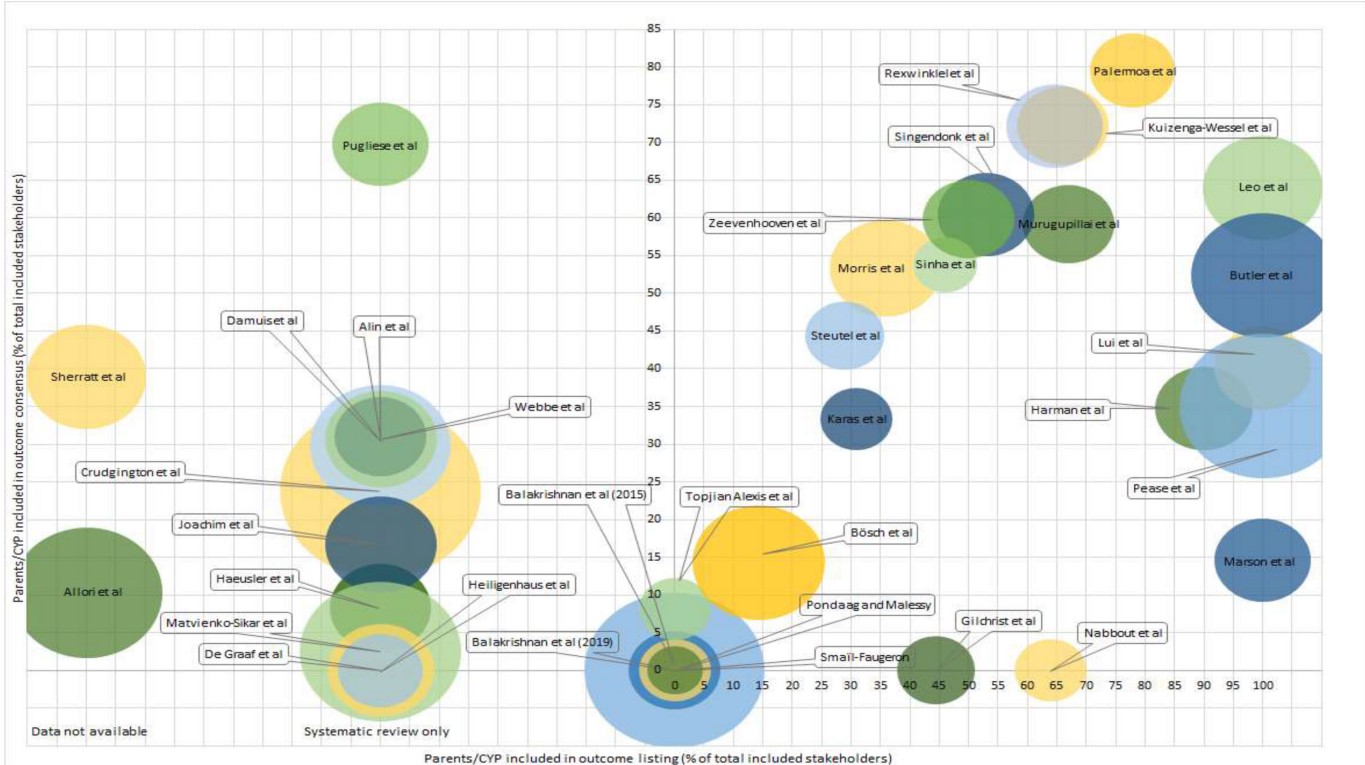

**Figure 4** Stakeholder involvement in outcome long listing and outcome consensus for all paediatric core outcome sets. Bubble size represents final paediatric core outcome set size. Studies to the left of the y-axis had no stakeholder involvement in the outcomes listing phase as outcome longlists were derived from systematic reviews, or the stakeholder breakdown was not published. CYP- children or young people.

was no compatible data to inform a joint, integrated guideline. Similar concerns apply to other major health areas of obesity, smoking and drinking, and uptake of vaccines.

Arguably, both of the public health and clinical approaches have emerged from measuring what can be measured within the current system and with existing instruments and been heavily influenced by the dominant expert paradigms of public health and clinical medicine. Both approaches can be criticised for overlooking important aspects of health and well-being valued by CYP, families and decision makers—and thus for resulting in data and findings with limited ability to inform decision-making. In the present review, we sought to adopt a third perspective to complement these two approaches, namely to identify ways forward for making more visible and measurable some of the important aspects of health that are not yet strongly featured in either of the existing approaches. Our findings here suggest that there indeed are shared, recognisable, universal health indicators that are likely to apply both to public health and clinical enquiries, such as self-care, pain, sleep and social interactions. These findings converge with, and further add to, the recent report from another international group that, independently and concurrently to us, sought to advance the thinking around child health measurement.[22] That concurrent work retained a healthcare paradigm and focused on existing standardised measurement tools

that could be used across diagnostic groups (explicitly excluding health indicators that were deemed not yet measurable through standardised instruments). Similarly to our review, they found universally important health indicators such as survival, growth, pain, school attendance and social functioning. Collectively, the findings from these two studies provide a strong foundation for the development of a universal, common child health indicator framework that spans traditional discipline and sector boundaries to complement existing core outcome sets for interventions as well as inform routine public health data collection. Developing such approach has the potential to facilitate more integrated, inclusive policy, practice and research across child health by focusing attention to universally important health and well-being goals that matter to all children, including children who may be clinically unwell.

To progress a meaningful child health indicator, framework will require further, substantial development. Key to this will be the involvement of a wider pool of stakeholders, particularly CYP, in deciding what the key indicators should be and how these are best operationalised for measurement and data collection. The findings from the present review align with those of the most recent COMET annual update[17] that found only 16% of paediatric core sets included direct input from CYP. While this is lower than 31% of sets found in this review, the authors included sets spanning both adult and CYP populations,

suggesting that with a broader review strategy CYP inclusion is even poorer. CYP and parents often differ in their priorities to health experts,[4 23 24] and it cannot be assumed that parents hold the same views as children and young people.[25] Therefore, including CYP as well as parents is crucial. There is evidence that CYP both understand and are keen to be involved in the development of health indicator and outcome sets[20] and in neonatal or infant conditions involvement of older children with relevant experiences may be useful. A key lesson from the present review is that those engaging higher numbers of CYP differed from standard Delphi methodology, perhaps paying particular attention to adapting the methods. Furthermore, focusing on a manageable number of indicators may facilitate stakeholder involvement.

There are three immediately actionable recommendations from the present study. First, national data set administrators should consider adding pain and sleep as key health indicators. Second, anyone developing core outcome or indicator sets should ensure they involve children, young people and parents—with adaptations to the methods to make this feasible. Third, clinical evaluators should consider inclusion of key, universally important child health outcomes that may be relevant to their interventions but absent from the current core sets.

**Contributors** VH and NK conceived of this project. VH developed the review protocol with input from NK and BA. VH and ER undertook title, abstract and full-text screening, and data extraction, and VH and NK undertook outcome coding. All coauthors provided input into the manuscript writing.

**Funding** Victoria Harbottle, HEE/NIHR ICA Predoctoral Fellow (NIHR300363), and Niina Kolehmainen, HEE/NIHR ICA Senior Clinical Lecturer (NIHR ICA-SCL-2015-01-003) were funded by the NIHR for this research project. The views expressed in this publication are those of the author(s) and not necessarily those of the NIHR, NHS or the UK Department of Health and Social Care.

**Competing interests** VH and NK were funded by the NIHR for part of this research project. No further competing interests to declare.

**Patient consent for publication** Not applicable.

**Ethics approval** Not applicable.

**Provenance and peer review** Not commissioned; externally peer reviewed.

**ORCID iDs**
Victoria Harbottle http://orcid.org/0000-0003-1731-464X
Chris Gale http://orcid.org/0000-0003-0707-876X
Niina Kolehmainen http://orcid.org/0000-0002-9229-9913

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
