## [Reviewer comments · BMJ Paediatrics Open]

ARTICLE DETAILS

TITLE (PROVISIONAL)	Identifying common health indicators from paediatric core outcome sets: a scoping review and mapping on the WHO International Classification of Functioning, Health and Disability
AUTHORS	Harbottle, Victoria Arnott, Bronia Gale, Chris Rowen, Elizabeth Kolehmainen, Niina

VERSION 1 – REVIEW

REVIEWER	Reviewer name: Doug Simkiss Institution and Country: University of Warwick, Ireland Competing interests: None
REVIEW RETURNED	05-Jun-2022

GENERAL COMMENTS	This is an important topic and I think the paper warrants publication. There are a few issues I would like the authors to consider: 1. Is there sufficient introductory information on Core Outcome Sets and the International Classification of Functioning, Disability and Health for non-specialists to understand these?2. In the search methodology the distinction between the COMET database search (Jan 2008 - March 2022) and the COMET database review (to February 2009) needs more explanation (though I did follow it in the end)3. Is it right to exclude Core Outcome Sets spanning children and adults? A sentence of two on this decision would assist and it could come up in a paragraph on limitations, which was absent.4. The second reviewer reviewed a proportion of abstracts / papers only, again this could be discussed in a paragraph on limitations5. Figure 3 needs a little more explanation in the text; do we assume all the papers had a stakeholder group as otherwise it is difficult to see how parents / children and young people could be lost from groups without PPI? If possible, confidence intervals on the bars would help interpretation of the figure6. Figure 4 is complex. I would suggest keeping the right half of the figure only and putting the 'data not available' and 'systematic review only' data on a separate figure afterwards. I really liked the emphasis on child, young person and family co-production and the discussion on public health and clinical versions on outcome data.
---

VERSION 1 – AUTHOR RESPONSE

Response to reviewers' comments

Reviewer Comment	Response
Editor in Chief "Table 2 would be better if the papers were listed in relation to disorders, eg gastrointestinal, neurological, etc"	Column for disorders added to table 2 and papers reordered
Associate Editor "My comments are less to do with the outcomes, as I think these are good and the paper worthy on this basis However, as a methodologist I do have some issues that need resolving....."	We have amended our description of the methodology accordingly
Dough Simkiss, University of Warwick "Is there sufficient introductory information on Core Outcome Sets and the International Classification of Functioning, Disability and Health for nonspecialists to understand these?"	Additional explanation added
"In the search methodology the distinction between the COMET database search (Jan 2008 - March 2022) and the COMET database review (to February 2009) needs more explanation (though I did follow it in the end)"	Sentence added to search methodology to clarify
"Is it right to exclude Core Outcome Sets spanning children and adults? A sentence of two on this decision would assist and it could come up in a paragraph on limitations, which was absent"	Limitations paragraph added to discussion
"The second reviewer reviewed a proportion of abstracts / papers only, again this could be discussed in a paragraph on limitations"	Limitations paragraph added to discussion
"Figure 3 needs a little more explanation in the text; do we assume all the papers 'Stakeholder group' refers to those included in the consensus rounds in the had a stakeholder group as otherwise it is difficult to see how parents / children and young people could be lost from groups without PPI?"	main studies and this figure looks to compare attrition across consensus rounds for studies that included PPI through advisory groups etc to those that did not. Clarifying text added
"If possible, confidence intervals on the bars would help interpretation of the figure 6"	Unsure what is meant as no figure 6 in paper
"Figure 4 is complex. I would suggest keeping the right half of the figure only and putting the 'data not available' and 'systematic review only' data on a separate figure afterwards"	We recognise that the figure is complicated. Before submission, and again as part of actioning the reviewer feedback, we have discussed and explored, in depth, various ways to visually display this information. We feel the real selling point of the current figure is that it allows visual comparison of everything in one place, and we feel this would be lost by splitting the figure, or cutting from it. Therefore, we would like to propose to retain the figure as it is; and have added a little more explanation to help the reader to navigate it.